# Daily disease severity in patients with COVID-19 admitted to the hospital: The SCODA (severity of coronavirus disease assessment) score

**Simone A. Joosten**[1☯], **Mark J. R. Smeets**[2☯]*, **M. Sesmu Arbous**[3], **Judith Manniën**[4], **Sander Laverman**[4], **Merijn M. G. Driessen**[4], **Suzanne C. Cannegieter**[2,5], **Anna H. E. Roukens**[1], on behalf of the Leiden University Medical Center BEAT-COVID group[¶]

**1** Department of Infectious Diseases, Leiden University Medical Center, Leiden, The Netherlands, **2** Department of Clinical Epidemiology, Leiden University Medical Center, Leiden, The Netherlands, **3** Department of Intensive Care, Leiden University Medical Center, Leiden, The Netherlands, **4** Department of Biomedical Data Sciences, Leiden University Medical Center, Leiden, The Netherlands, **5** Department of Internal Medicine, Section Thrombosis and Hemostasis, Leiden University Medical Center, Leiden, The Netherlands

☯ These authors contributed equally to this work.
¶ Membership of the Leiden University Medical Center BEAT-COVID group is provided in the Acknowledgments.
* m.j.r.smeets@lumc.nl

**Data Availability Statement:** All data used to create Figs 2, 3 and 4 are included in the Supporting Information files. Data used for Table 2

## Abstract

### Background

A multitude of diagnostic and predictive algorithms have been designed for COVID-19. However, currently no score can accurately quantify and track day-to-day disease severity in hospitalised patients with COVID-19. We aimed to design such a score to improve pathophysiological insight in COVID-19.

### Methods

Development of the Severity of COronavirus Disease Assessment (SCODA) score was based on the 4C Mortality score but patient demographic variables that remain constant during admission were excluded. Instead, parameters associated with breathing and oxygenation were added to reflect the daily condition. The SCODA score was subsequently applied to the BEAT-COVID cohort to describe COVID-19 severity over time and to determine the timing of clinical recovery for each patient, an important marker in pathophysiological studies. The BEAT-COVID study included patients with PCR confirmed COVID-19 who were hospitalized between April 2020 and March 2021 in the Leiden University Medical Center, The Netherlands.

### Results

The SCODA score consists of 6 clinical and 2 routine lab parameters. 191 patients participated in the BEAT-COVID study. Median age was 66, and 74.4% was male. The modal

cannot be shared publicly because of potential patient identification risks. Requests to obtain these data can be addressed to the LuCID Scientific Committee (contact via Lucid_weco@lumc.nl).

**Funding:** This project was partly supported by Crowdfunding Wake Up To Corona, organized by the Leiden University Fund. The funder had no role in study design, data collection and analysis, decision to publish, or preparation of the manuscript.

**Competing interests:** The authors have declared that no competing interests exist.

timepoint at which recovery was clinically initiated occurred on days 8 and 24 since symptom onset for non-ICU and ICU-patients, respectively.

## Conclusions

We developed a daily score which can be used to track disease severity of patients admitted due to COVID-19. This score is useful for improving insight in COVID-19 pathophysiology, its clinical course and to evaluate interventions. In a future stage this score can also be used in other (emerging) infectious respiratory diseases.

## Introduction

Coronavirus disease (COVID-19) is caused by the novel SARS-CoV-2 coronavirus that was first identified in Wuhan, Hubei, China in December 2019 [1]. Infection with SARS-CoV-2 can result in severe viral pneumonia in which the most severe cases deteriorate and develop acute respiratory distress syndrome, requiring prolonged mechanical ventilation [2]. In March 2020, the WHO declared COVID-19 a global pandemic. Five million deaths resulted directly from the pandemic in the first 18 months and hence research has been directed at finding ways to decrease the mortality and morbidity of COVID-19 [3]. Intriguingly, inter-individual heterogeneity in disease severity is large. Symptoms of an infection can range from very mild and short lived to severe with requirement of mechanical ventilation for multiple weeks, without clear understanding of this divergency [4, 5]. To better understand the pathophysiology of an acute infectious respiratory disease (such as COVID-19), its clinical course and to evaluate therapeutic interventions, a detailed daily assessment of the disease severity is necessary.

Thorough longitudinal analyses on patients admitted to the hospital with COVID-19, such as detailed immune monitoring studies, would greatly benefit from objective daily insights in the patients' clinical condition. Daily scoring of disease severity, with objective and clinically available parameters, will also permit to determine clinical deterioration or improvement at a higher resolution. In addition, it will permit to correlate biomarkers to actual clinical states, thus increasing mechanistic and pathophysiological insight which could enable the identification of predictive entities for future immune monitoring and intervention studies.

Currently, most available illness scores for hospital admitted patients, such as the APACHE (acute physiology and chronic health evaluation) score or the 4C Mortality score, are predictive for clinical outcomes [6, 7]. However, during hospital admission, patients experience multiple phases of different disease severity which are poorly reflected by these classical predictive scores. Many variables that compose the basis of the predictive scores, such as age, sex and comorbidities, do not change during hospital admission and may thus strongly dampen the variables that reflect the actual ongoing disease processes.

An example of a valuable daily score to identify a critically ill patient on the general ward is the MEWS (Modified Early Warning Score) [8]. The MEWS is, however, not applicable to ICU admitted patients as MEWS-parameters respiratory rate and consciousness are not reflections of the patients' state due to sedation and mechanical ventilation in patients admitted to the Intensive Care Unit (ICU). Besides respiratory variables, the MEWS focuses on hemodynamic variables which are less relevant in COVID-19. Therefore, the MEWS is not suitable to track disease severity of patients with COVID-19 or other patients with infectious respiratory diseases both on the ward and ICU.

As there are currently no disease severity scores which allow to assess the daily status of patients with COVID-19, we developed the 'severity of coronavirus disease assessment' (SCODA) score. The aim for the development of the SCODA score was multifold:

First, to assess the severity of COVID-19 related illness, based on objective, straightforward and routinely clinically available parameters on a daily basis. Such a severity score would be useful in assessing disease severity status of patients hospitalized for COVID-19 at a given time (cross sectional) and in tracking the course of the disease over time (longitudinal).

Secondly, to develop a score that could be applied to both patients at the general ward (non-ICU) and patients in the ICU, which is key since patients with COVID-19 are often transferred back and forth between these two departments.

Thirdly, to call for a more-distinctive characterization of disease severity in patients hospitalized for COVID-19, which would enable detection of small to moderate improvements or deteriorations when therapeutic interventions are done.

Fourthly, to be able to link COVID-19 related illness to laboratory (bio)markers in a more detailed manner than categorizing patients as outpatient, ward or ICU patients. Variations in clinical severity may very well reflect underlying pathophysiological processes related to inflammation, which may be more easily identified by linking them to a daily disease severity score.

And finally, to be able to define, based on this daily severity score, a recovery point for each patient, which is the time point during hospital admission after which the patient is clearly recovering and does not deteriorate anymore. This would enable biomarker analysis to be correlated to deterioration and recovery, which is more representative of the biological processes than categorizing patients merely in non-ICU or ICU-patients.

After construction of the SCODA score it was applied to the BEAT-COVID study cohort to describe COVID-19 severity over time and to determine the timing of clinical recovery for each patient.

## Methods

### SCODA score development

To create a COVID-19 severity score that would be easy to use and directly available, a limited number of straightforward, objective and easily obtainable variables that are routinely measured and registered as part of clinical care in patients with COVID-19 were included.

The 4C Mortality score was used as the basis for development of the SCODA score as many independently measured parameters associated with respiration, infection and organ function are included in this score [6]. However, since the SCODA score aimed to reflect daily disease severity rather than mortality risk, patient demographic variables were removed (i.e., age, sex and comorbidities at admission). Instead, as COVID-19 disease severity is mostly associated with insufficient oxygenation multiple, independently determined, and routinely recorded parameters associated with breathing and oxygenation were included in the design, which account for 10 out of the maximum score of 17. These parameters are oxygen flow (l/min) and saturation (%) for patients not admitted to the ICU, and PaO2/FiO2 (P/F) ratio (mmHg or kPa) for patients admitted to the ICU. The P/F ratio is a well-established tool to identify hypoxemic respiratory failure [7]. As the relation between P/F and FiO2 is neither constant nor linear, even when shunt remains constant, we also included the FiO2 (% or fraction) as a separate parameter in the score for ICU patients [9]. Ultimately, the SCODA score consisted of the following parameters: respiratory rate, peripheral oxygen saturation on room air, P/F Ratio, oxygen flow, FiO2, Glasgow Coma Scale (GCS), blood urea level and C-reactive protein (CRP) (**Table 1**).

**Table 1. Variables included in the Severity of COronavirus Disease Assessment (SCODA) score.**

| Variable | SCODA score | |
|---|---|---|
| | Non-ICU (0–15) | ICU (0–17) |
| **Respiratory rate (breaths/min)** | | |
| <20 | 0 | 0 |
| 20–29 | 1 | 1 |
| ≥30 | 2 | 2 |
| mechanical ventilation | | 3 |
| **Peripheral oxygen saturation on room air (%)[1]** | | |
| <92 | 2 | |
| ≥92 | 0 | |
| **P/F ratio[2]** | | |
| <100 mmHg / <13.3 kPa | | 3 |
| 100–199 / 13.3–26.6 kPa | | 2 |
| 200–299 / 26.6–39.9 kPa | | 1 |
| ≥300 / ≥39.9 kPa | | 0 |
| **Oxygen flow (l/min)[1]** | | |
| <1 | 0 | |
| 1–2 | 1 | |
| 3–5 | 2 | |
| 6–8 | 3 | |
| ≥9 | 4 | |
| **FiO2 (%)[2]** | | |
| <30 | | 0 |
| 30–44 | | 1 |
| 45–59 | | 2 |
| 60–79 | | 3 |
| ≥80 | | 4 |
| **Glasgow Coma Scale score** | | |
| <15 | 2 | 2 |
| 15 | 0 | 0 |
| **Urea (mmol/L)** | | |
| ≤7 | 0 | 0 |
| 8–14 | 1 | 1 |
| >14 | 3 | 3 |
| **C Reactive Protein (mg/L)** | | |
| <50 | 0 | 0 |
| 50–99 | 1 | 1 |
| > 99 | 2 | 2 |

1 only during admittance to the ward (non-ICU)
2 only during admittance to the ICU

Parameter scores were based on the 4C Mortality score and clinical expertise (e.g., cut-off values that prompt clinicians to re-evaluate the clinical state of the patient often with consequences for treatment). To verify that changes in the SCODA score over time reflected the changes in disease severity for individual patients, an Infectious Diseases specialist and an Intensivist both assessed the score in 15 electronic patient files each (patients not admitted to the ICU for the Infectious Diseases specialist and admitted to the ICU for the Intensivist).

In clinical practise, respiratory rates, serum CRP, FiO2 and blood urea levels may be measured multiple times a day. For the SCODA score, the highest daily value for each of these should be used as it reflects the worst clinical state. Similarly, for P/F ratio and the GCS, the lowest daily value should be used, also reflecting the worst state. Regarding oxygen flow and peripheral oxygen saturation on the ward, which can be highly variable during one day, the daily average value should be used for the SCODA score.

## Application of SCODA score in BEAT-COVID cohort

**Population.** The SCODA score was applied to the BEAT-COVID cohort which includes 191 participants, diagnosed with PCR positive COVID-19, who required hospital admission at the Leiden University Medical Center, Leiden, the Netherlands, between April 2020 and March 2021 (**Fig 1**). In this population of patients admitted to the hospital (both non-ICU and ICU), treatment was initially limited to supportive care only, though patients admitted after August 2020 also received dexamethasone. At the time of inclusion, anti-IL6R was only used in some cases at our centre, anti-SARS-CoV-2 monoclonal antibodies had not yet been investigated, and no COVID-19 vaccines had been administered to non-healthcare workers at that time in the Netherlands yet. Importantly, the BEAT-COVID study was conducted before development of the SCODA score and hence, measurements in this cohort were not influenced by the score.

**Patient consent statement.** All patients or their close relatives provided written informed consent to participate in the BEAT-COVID study. The principal investigator had access to information with which to identify individual patients.

**Ethical approval.** Ethical approval for the study protocol was provided by the Leiden University Medical Center Medical Ethics committee (protocol NL73740.058.20). The trial was

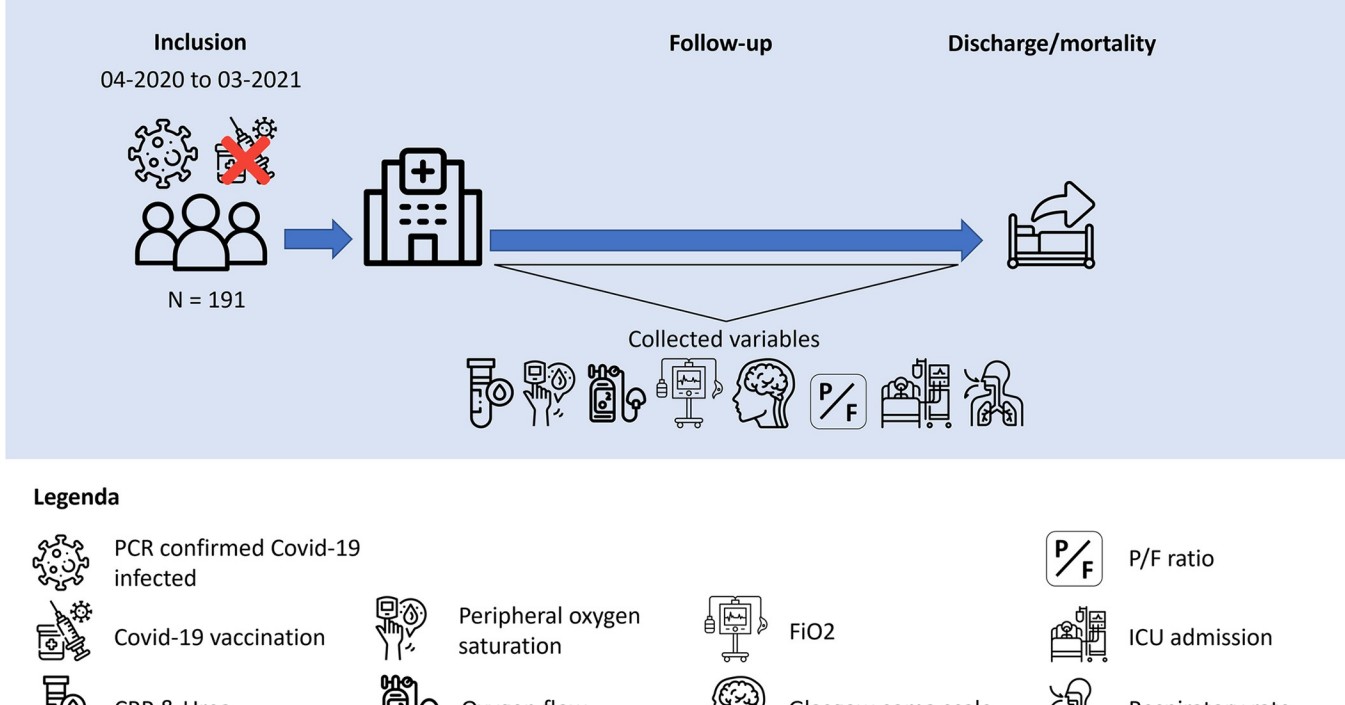

**Fig 1. Visual explanation of the inclusion, follow-up and data collection in the BEAT-COVID cohort.**

registered at the International Clinical Trials Registry Platform (https://trialsearch.who.int/Trial2.aspx?TrialID=NL8589).

**Score measurements.** A daily SCODA score was retrospectively calculated in all patients for the duration of hospital admission. Since the BEAT-COVID population was not constructed for the SCODA score, information on the score's parameters was not available for every day of admission. The GCS for example in patients admitted to the ward was only scored if patients had signs of altered mental state. Therefore, patients admitted to the ward without registered GCS score were assumed to have the maximum score of 15 points, which reflects an unaltered mental state. Similarly, if oxygen flow was not reported, it was assumed that patients had no need for it. Furthermore, missing data on respiratory rate, oxygen saturation, P/R ratio, FiO2, urea and CRP was handled by linear interpolation. For parameters for which missing data could not be interpolated, a score of 0 was given (except for GCS as explained above) since missingness was likely due to a better clinical condition which did not warrant lab or clinical measurements.

**COVID-19 severity over time.** The development of disease severity was assessed over time for each patient since the self-reported date of disease onset. We chose days since onset of symptoms instead of days since hospital admittance, as days since symptoms onset reflect more accurately the pathophysiological course of the disease than days since hospitalisation. Namely, the timing of hospital admittance can be influenced by several factors such as patients or doctor's delay, or the decision of the emergency ward doctor to admit a patient which are culturally determined or calendar time-dependent.

**Recovery timepoint.** To understand pathophysiological processes behind disease and recovery it is critical to determine the time point where recovery set in. The recovery time point was per definition after the highest severity score for that individual patient and was defined as a SCODA score of 5 or below for at least two consecutive days. Furthermore, after the recovery start time only one SCODA score of 7 or above might have occurred at later time points. This was allowed since several patients had a temporal decline in clinical condition after transport from the ICU to the ward, e.g., requiring temporarily more oxygen. The recovery time point was assessed for all patients included in the BEAT-COVID study that recovered during their hospital stay, i.e., when outcome was not death, using the rules described above. In fifteen randomly selected patients, the recovery point was visually verified by the supervising Infectious Diseases consultant in the individual SCODA score plots.

## Statistical analyses

Baseline characteristics were reported as median with interquartile range or numbers with proportions, as appropriate.

For the visual representation of the COVID-19 severity over time, we plotted the SCODA scores versus the days since onset of disease and fitted a smoothed line with 95% confidence interval. Similarly, to visualise the recovery time points in the BEAT-COVID cohort, we plotted the distribution of recovery days, split by whether a patients had been admitted to the ICU at any timepoint during his/her admission. Lastly, we reported the modal (i.e., most frequent) recovery timepoint, again split by whether a patients had been admitted to the ICU or not at any point during admission.

## Results

### Population

The median age of the study population was 66 years (interquartile range [IQR] 58–73), 74.4% was male and the median BMI on admission was 28.4 kg/m2 (IQR 25.1–31.4 kg/m2) (**Table 2**).

**Table 2. Characteristics of the BEAT-COVID study population (N = 191).**

| Characteristic | N = 191 | Missing |
|---|---|---|
| Sex (Male) | 142 (74.4%) | - |
| Age (years) | 66 (median), 58–73 (IQR) | - |
| BMI at admission (kg/m²) | 28.4 (median), 25.1–31.4 (IQR) | 13 |
| ICU admission | 104 (46%) | |
| Discharged alive | 159 (83.3%) | - |
| Duration of hospital admission (days) | 9 (median), range 2–145 days | |
| Comorbidities (excl. BMI) | | |
| Patients without comorbidities | 51 (26.7%) | |
| Patients with 1 comorbidity | 56 (29.3%) | |
| Patients with more than 1 comorbidity | 84 (44.0%) | |
| *Chronic cardiac disease* | 60 (31.4%) | 1 |
| *Hypertension* | 75 (39.3%) | - |
| *Chronic Pulmonary Disease* | 17 (8.9%) | 1 |
| *Asthma* | 28 (14.7%) | 1 |
| *Chronic Neurological Disease* | 15 (7.9%) | 2 |
| *Diabetes Mellitus* | 62 (32.5%) | - |
| *Chronic Liver Disease* | 5 (2.6%) | 1 |
| *Chronic Kidney Disease* | 17 (8.9%) | 1 |

## SCODA score application in BEAT-COVID cohort

For the total BEAT-COVID population, the maximum severity score, i.e., 15 points for ward patients and 17 for ICU patients, was observed typically between days 10 and 40 after symptom onset (**Fig 2**). The highest mean severity score was observed around day 30 after symptom onset, indicating that in the hospitalised population, most people experienced the worst stage of disease around 30 days after symptom onset. In our population, the mean SCODA score eventually did not become zero, as 17% of patients died at the hospital, and 14% was transferred to another hospital or a rehabilitation clinic with remaining symptoms.

## Recovery timepoint

By plotting the scores of individual patients, it was observed that the SCODA score follows a sawtooth pattern for patients admitted to the ICU, which is reflecting also the subtle clinical changes, and a somewhat smoother line for patients not admitted to the ICU as exemplified for four patients in **Fig 3**. The recovery timepoints based on the SCODA score corresponded well with the clinical course as assessed by the Infectious Diseases consultant. The modal recovery timepoint for patients never admitted to the ICU was day 8 since symptom onset while this was day 24 for patients who were admitted to the ICU at any point during their hospital admission (**Fig 4**).

## Discussion

A daily severity score of patients with COVID-19, based on the 4C Mortality score and clinical experience which includes common, clinically available data was designed. Alterations of the 4C Mortality score, based on clinical expertise, were needed to reflect the aim of the SCODA score to classify the disease severity at a time point as opposed to the prediction of clinical outcome (e.g., mortality for the 4C Mortality score). We found that this hands-on score reflects the actual daily severity of patients with COVID-19 well. Furthermore, we found that, in the

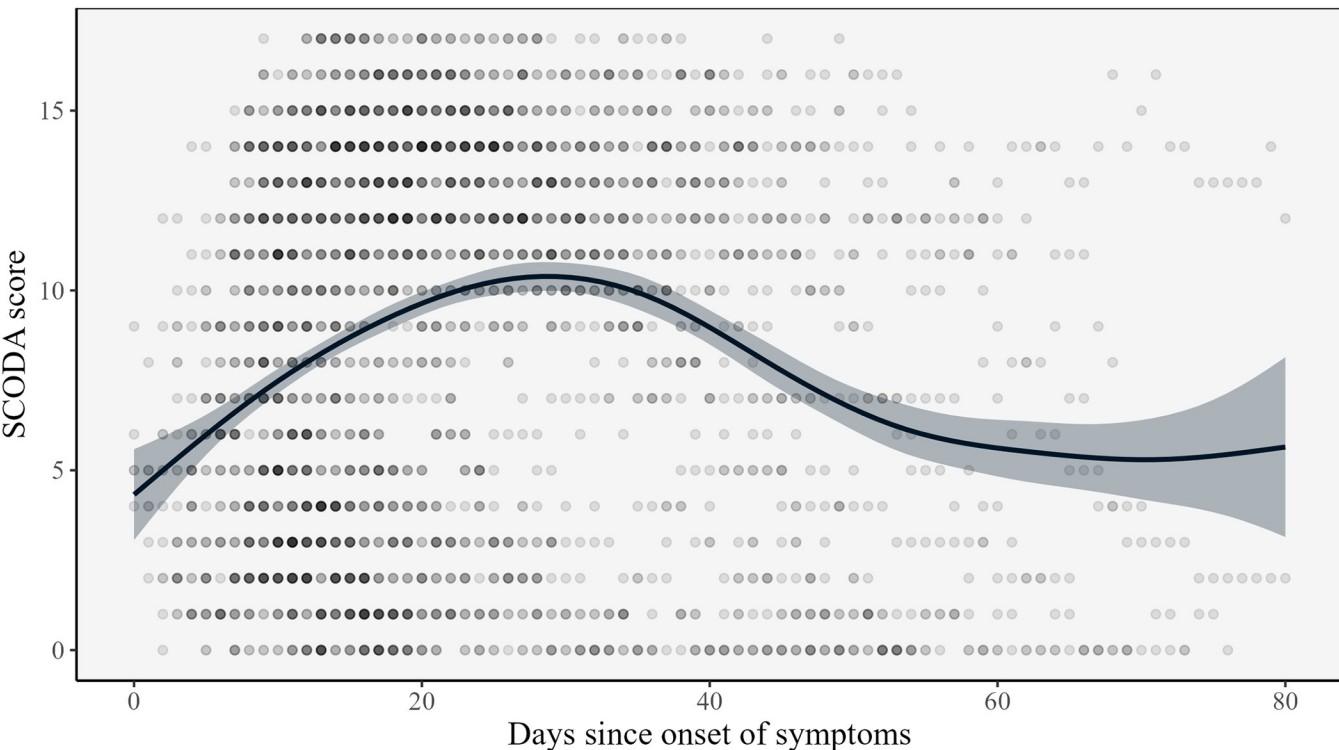

**Fig 2. Correlation between SCODA score and days since symptom onset in patients with COVID-19 admitted to the Leiden University Medical Center in the Netherlands (the BEAT-COVID population).** Daily SCODA scores of patients are shown as dots, with the darker dots indicating more overlapping data points. Patients have a score for every single day of hospitalisation. The line represents the mean SCODA score per day since onset of symptoms. The grey area is the 95% confidence interval. Outliers with a day of recovery greater than 80 were removed from the analysis to improve internal validity.

BEAT-COVID population, on average patients experienced their worst disease severity around day 30 after start of symptoms while the modal for the timepoint of recovery was 8 days and 24 days for patients that were and weren't admitted to the ICU, respectively. At first sight these findings might seem contradictory. However, this is possible as patients that died contributed to the estimation of the mean SCODA score over time but, since they died, they did not have a timepoint of recovery.

Currently, though many diagnostic and prognostic models have been developed for COVID-19, there is no comprehensive score which can be used to track the disease severity of patients with COVID-19 over time as it is, so unrelated to any outcome and not with the aim to do so [10]. Multiple studies which investigated COVID-19 have relied on less precise classifications of disease severity (i.e., classification based on overall symptom severity, any admittance to the ICU or outcome) to categorize patients [11–14]. This broad categorizations limits investigations of, for example, relationships between biomarkers and COVID-19 severity as both will fluctuate over the course of the disease. This is precisely were the SCODA score adds value as it allows for day-to-day monitoring of the disease severity status which can then be correlated to biomarkers. These correlations with biomarkers can subsequently lead to new targets for monitoring of interventions or for developing and updating prediction models.

A point of consideration in using the SCODA score is that, while the SCODA score was found to reflect the disease severity in the BEAT-COVID cohort well, this cohort consisted of patients treated in the LUMC and were therefore subject to the same clinical expertise that is incorporated in the SCODA score. A limitation of this is that, while the SCODA score may

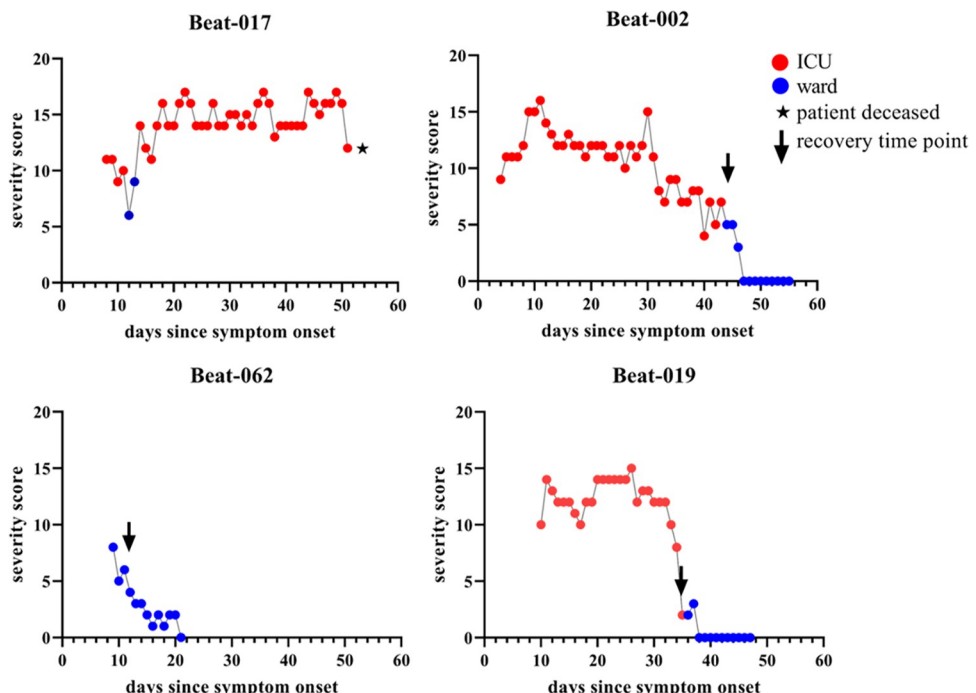

**Fig 3. COVID-19 severity trajectories in four patients.** One graph represents one individual patient. Red dots represent the SCODA score while the patient was admitted to the ICU, blue lines represent the score while the patient was admitted to the ward. Recovery time points were determined for these patients, patient BEAT-017 deceased and thus had no recovery time point, patient BEAT-002 recovered from day 44, BEAT-019 from day 35 and BEAT-062 from day 12.

also contribute to important insights in pathophysiological processes in other cohorts, the score should be verified in these populations before application. This validation is needed because the SCODA score is developed to capture disease severity, something for which there is no golden standard. Therefore, the main uncertainty in using the SCODA score in other settings is whether the interpretation of disease severity, as quantified by the SCODA score, reflects the subjective interpretation of the clinicians. This is why the SCODA score should be validated by new researchers and with each successful validation the likelihood that the score reflects the disease severity accurately increases. Hence, we encourage clinicians and researchers to validate the score and to subsequently incorporate the SCODA score in their immunological research to increase insight in underlying pathophysiology in deteriorating and recovering of patients with COVID-19.

Another point of consideration is that, while interpreting the estimated SCODA score measurements, it is necessary to take into account temporal factors, such as the gradual change over time in dominant COVID-19 variants (during inclusion of the BEAT-COVID population this was the Wuhan variant), public awareness and treatment protocols. In the first wave for example, patients were admitted to the hospital at a later time point after disease onset in general than patients during the second wave, partly due to hospital bed availability and not related to degree of illness. Hence, the timepoint at which patients experience the worst stage of disease, but also the recovery point, might differ substantially depending the afore mentioned factors. So this limitation mainly applies to the question of generalisability of the (aggregated) measurements in our BEAT-COVID cohort. The SCODA score itself will likely not be affected by these temporal factors as the variables included in the score are independent from

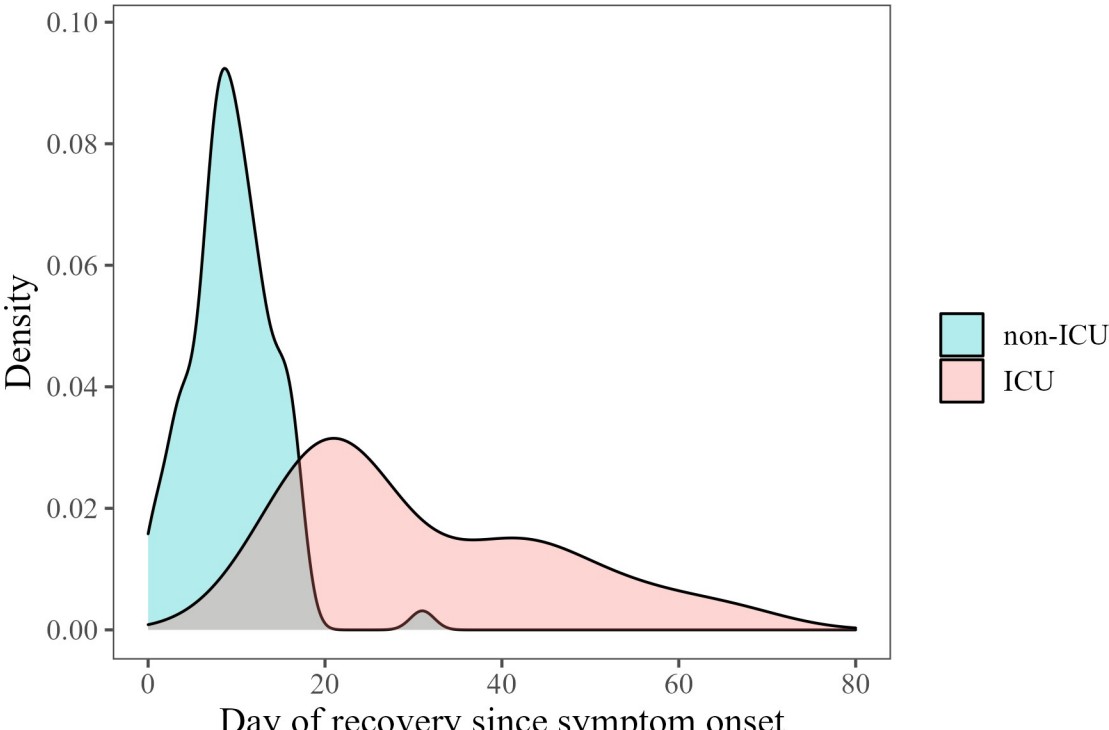

**Fig 4. Distribution of recovery timepoints for patients who survived and where either never admitted to the ICU (non-ICU) or were admitted to ICU at any point during their admission (ICU).** For patients not admitted to the ICU the modal timepoint of recovery was day 8 while for patients admitted to the ICU at any time during hospital admission this was on day 24. Outliers with a day of recovery greater than 80 were removed to improve internal validity.

these factors. For example, when a patient is vaccinated but still gets severely ill with high CRP, the need for mechanical ventilation and a decreased GCS, his/her disease severity will still be accurately quantified despite the vaccination status. Only when new COVID-19 variants lead to different symptom patterns (e.g., severe gastro-intestinal or urinary tract symptoms without concomitant respiratory symptoms) will there be a mismatch in the SCODA score and the actual disease severity.

Ultimately, we have developed a score which can be used to, cross-sectionally or longitudinally, assess daily severity of COVID-19 and combine this with for example laboratory (bio) markers to unravel underlying pathophysiological processes as these are related to clinical severity. In a future stage, possibly with minor modifications, the SCODA score might also be useful in tracking the disease severity of other known (e.g., influenza) and possible novel infectious respiratory diseases.

## Supporting information

**S1 Data.**
(XLSX)

**S2 Data.**
(XLSX)

## Acknowledgments

We want to thank the department of Biomedical Data Sciences (LUMC) for their support in data management, and the department of Clinical Epidemiology (LUMC) for their clinical COVID-19 data collection.

Leiden University Medical Center BEAT-COVID group (in alphabetical order)

M. Sesmu Arbous[1], Bernard M. van den Berg[2], Suzanne Cannegieter[3], Christa M. Cobbaert[4], Anne M. van der Does[5], Jacques J.M. van Dongen[6], Jeroen Eikenboom[7], Mariet C.W. Feltkamp[8], Annemieke Geluk[9], Jelle J. Goeman[10], Martin Giera[11], Thomas Hankemeier[12], Mirjam H.M. Heemskerk[13], Pieter S. Hiemstra[5], Cornelis H. Hokke[14], Jacqueline J. Janse[14], Simon P. Jochems[14], Simone A. Joosten[9], Marjolein Kikkert[8], Lieke Lamont[12], Judith Manniën[10], Tom H.M. Ottenhoff[9], T. Pongracz[11], Michael R. del Prado[1], Meta Roestenberg[9,14], Anna H.E. Roukens[9,#], Hermelijn H. Smits[14], Eric J. Snijder[8], Frank J.T. Staal[6], Leendert A. Trouw[6], Roula Tsonaka[10], Aswin Verhoeven[11], Leo G. Visser[9], Jutte J.C. de Vries[8], David J. van Westerloo[1], Jeanette Wigbers[1], Henk J. van der Wijk[10], Robin C. van Wissen[4], Manfred Wuhrer[11], Maria Yazdanbakhsh[14], Mihaela Zlei[6]

Affiliations

1. Department of Intensive Care, Leiden University Medical Center, Leiden, the Netherlands

2. Department of Internal Medicine, Leiden University Medical Center, Leiden, the Netherlands

3. Department of Clinical Epidemiology, Leiden University Medical Center, Leiden, the Netherlands

4. Department of Clinical Chemistry, Leiden University Medical Center, Leiden, the Netherlands

5. Department of Pulmonology, Leiden University Medical Center, Leiden, the Netherlands

6. Department of Immunology, Leiden University Medical Center, Leiden, the Netherlands

7. Department of Internal Medicine, Leiden University Medical Center, Leiden, the Netherlands

8. Department of Medical Microbiology, Leiden University Medical Center, Leiden, the Netherlands

9. Department of Infectious Diseases, Leiden University Medical Center, Leiden, the Netherlands

10. Department of Biomedical Data Sciences, Leiden University Medical Center, Leiden, the Netherlands

11. Center for Proteomics and Metabolomics, Leiden University Medical Center, Leiden, the Netherlands

12. Division of Systems Biomedicine and Pharmacology, Leiden Academic Center for Drug Research, Leiden University, the Netherlands

13. Department of Hematology, Leiden University Medical Center, Leiden, the Netherlands

14. Department of Parasitology, Leiden University Medical Center, Leiden, the Netherlands

# Principal investigator and lead author (A.H.E. Roukens), email: A.H.E.Roukens@lumc.nl

## Author Contributions

**Conceptualization:** Simone A. Joosten, M. Sesmu Arbous, Suzanne C. Cannegieter, Anna H. E. Roukens.

**Data curation:** Mark J. R. Smeets, Judith Manniën, Sander Laverman, Merijn M. G. Driessen.

**Formal analysis:** Simone A. Joosten, Mark J. R. Smeets, Sander Laverman, Merijn M. G. Driessen.

**Funding acquisition:** Simone A. Joosten, Anna H. E. Roukens.

**Methodology:** Simone A. Joosten, Mark J. R. Smeets, M. Sesmu Arbous, Suzanne C. Cannegieter, Anna H. E. Roukens.

**Project administration:** Judith Manniën, Anna H. E. Roukens.

**Supervision:** Simone A. Joosten, M. Sesmu Arbous, Suzanne C. Cannegieter, Anna H. E. Roukens.

**Visualization:** Mark J. R. Smeets, Sander Laverman.

**Writing – original draft:** Simone A. Joosten, Mark J. R. Smeets.

**Writing – review & editing:** Simone A. Joosten, M. Sesmu Arbous, Judith Manniën, Suzanne C. Cannegieter, Anna H. E. Roukens.

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
