## [Decision Letter · Decision Letter 0]

6 Jul 2023

PONE-D-23-08857Daily disease severity in patients with COVID-19 admitted to the hospital: the SCODA (severity of coronavirus disease assessment) scorePLOS ONE

Dear Dr. Smeets,

Thank you for submitting your manuscript to PLOS ONE. After careful consideration, we feel that it has merit but does not fully meet PLOS ONE’s publication criteria as it currently stands. Therefore, we invite you to submit a revised version of the manuscript that addresses the points raised during the review process. 

Dear Dr. Mark Smeets,

Thank you for submitting your manuscript to Australian Critical Care.

We are interested in publishing your work, but we believe it requires more fully addressing major and minor comments before reconsideration for publication in the journal. My comments are included below for your attention.

1.Please clrealy state Ethical approval in Ethics part.

2.Make "statistical" section in Method.

Academic Editor Dr. Yujiro Matsuishi

Reveiwer 1

Minor comment:Dear authors, many thanks for your work. Its interesting and globally needed topic. I recommend drawing a diagram to illustrate the method section and data collection.

Reveiwer 2:

The discussion section provides a comprehensive overview of the SCODA score and its potential applications in tracking the disease severity of patients with COVID-19. Here are a few suggestions for improvement:

1. It would be helpful to briefly explain how the SCODA score was developed, including the rationale for using the 4C Mortality score and incorporating clinical experience. This would give readers a better understanding of the score's foundation and its potential advantages over other existing scores or classifications.

2. While the discussion mentions that there is currently no comprehensive score available for tracking the disease severity of COVID-19 patients over time, it would be beneficial to elaborate on the significance of this gap in knowledge. Discuss how the SCODA score fills this gap and how it can provide valuable insights into disease progression and its correlation with biomarkers.

3. Further elaborate on the point regarding the potential impact of temporal factors, such as the change in dominant COVID-19 variants, public awareness, and treatment protocols, on the interpretation of the SCODA score measurements. Discuss how these factors may influence the timing of disease severity and recovery, and acknowledge their potential influence on the generalizability of the findings.

4. Although the discussion mentions the need for validation and acknowledges potential limitations of the SCODA score, it would be helpful to explicitly state these limitations. Discuss any inherent biases or uncertainties associated with the score, as well as any potential sources of error or confounding factors that may affect its accuracy or applicability.

We look forward to receiving your revised manuscript.

Kind regards,

Yujiro Matsuishi

Academic Editor

PLOS ONE

3. One of the noted authors is a group or consortium [Leiden University Medical Center BEAT-COVID group]. In addition to naming the author group, please list the individual authors and affiliations within this group in the acknowledgments section of your manuscript. Please also indicate clearly a lead author for this group along with a contact email address.

Additional Editor Comments:

Dear Dr. Mark Smeets,    

Thank you for submitting your manuscript to Australian Critical Care.

We are interested in publishing your work, but we believe it requires more fully addressing major and minor comments before reconsideration for publication in the journal. My comments are included below for your attention.

1.Please clrealy state Ethical approval in Ethics part.

2.Make "statistical" section in Method.

Reveiwer 1

Minor comment:Dear authors, many thanks for your work. Its interesting and globally needed topic. I recommend drawing a diagram to illustrate the method section and data collection.

Reveiwer 2:

The discussion section provides a comprehensive overview of the SCODA score and its potential applications in tracking the disease severity of patients with COVID-19. Here are a few suggestions for improvement:

1. It would be helpful to briefly explain how the SCODA score was developed, including the rationale for using the 4C Mortality score and incorporating clinical experience. This would give readers a better understanding of the score's foundation and its potential advantages over other existing scores or classifications.

2. While the discussion mentions that there is currently no comprehensive score available for tracking the disease severity of COVID-19 patients over time, it would be beneficial to elaborate on the significance of this gap in knowledge. Discuss how the SCODA score fills this gap and how it can provide valuable insights into disease progression and its correlation with biomarkers.

3. Further elaborate on the point regarding the potential impact of temporal factors, such as the change in dominant COVID-19 variants, public awareness, and treatment protocols, on the interpretation of the SCODA score measurements. Discuss how these factors may influence the timing of disease severity and recovery, and acknowledge their potential influence on the generalizability of the findings.

4. Although the discussion mentions the need for validation and acknowledges potential limitations of the SCODA score, it would be helpful to explicitly state these limitations. Discuss any inherent biases or uncertainties associated with the score, as well as any potential sources of error or confounding factors that may affect its accuracy or applicability.

Reviewers' comments:

Reviewer's Responses to Questions

**Comments to the Author**

1. Is the manuscript technically sound, and do the data support the conclusions?

Reviewer #1: Yes

Reviewer #2: Yes

2. Has the statistical analysis been performed appropriately and rigorously? 

Reviewer #1: Yes

Reviewer #2: Yes

3. Have the authors made all data underlying the findings in their manuscript fully available?

Reviewer #1: Yes

Reviewer #2: Yes

4. Is the manuscript presented in an intelligible fashion and written in standard English?

Reviewer #1: Yes

Reviewer #2: Yes

5. Review Comments to the Author

Reviewer #1: Dear authors, many thanks for your work. Its interesting and globally needed topic. I recommend drawing a diagram to illustrate the method section and data collection.

Regards

Reviewer #2: The discussion section provides a comprehensive overview of the SCODA score and its potential applications in tracking the disease severity of patients with COVID-19. Here are a few suggestions for improvement:

1. It would be helpful to briefly explain how the SCODA score was developed, including the rationale for using the 4C Mortality score and incorporating clinical experience. This would give readers a better understanding of the score's foundation and its potential advantages over other existing scores or classifications.

2. While the discussion mentions that there is currently no comprehensive score available for tracking the disease severity of COVID-19 patients over time, it would be beneficial to elaborate on the significance of this gap in knowledge. Discuss how the SCODA score fills this gap and how it can provide valuable insights into disease progression and its correlation with biomarkers.

3. Further elaborate on the point regarding the potential impact of temporal factors, such as the change in dominant COVID-19 variants, public awareness, and treatment protocols, on the interpretation of the SCODA score measurements. Discuss how these factors may influence the timing of disease severity and recovery, and acknowledge their potential influence on the generalizability of the findings.

4. Although the discussion mentions the need for validation and acknowledges potential limitations of the SCODA score, it would be helpful to explicitly state these limitations. Discuss any inherent biases or uncertainties associated with the score, as well as any potential sources of error or confounding factors that may affect its accuracy or applicability.

6. PLOS authors have the option to publish the peer review history of their article (what does this mean?). If published, this will include your full peer review and any attached files.

Reviewer #1: No

Reviewer #2: No

---

## [Author Response · Author response to Decision Letter 0]

12 Jul 2023

Comments by the Academic Editor:

1. Please clearly state Ethical approval in Ethics part.

We have separated the short paragraph on the ethical approval by the Leiden University Medical Center Medical Ethics committee as separate paragraph of the methods section. 

2.Make "statistical" section in Method.

We have included the statistical section below to the method section:

(P10 L17-27)

Statistical analyses

Baseline characteristics were reported as median with interquartile range or numbers with proportions, as appropriate. 

For the visual representation of the COVID-19 severity over time, we plotted the SCODA scores versus the days since onset of disease and fitted a smoothed line with 95% confidence interval. Similarly, to visualise the recovery time points in the BEAT-COVID cohort, we plotted the distribution of recovery days, split by whether a patients had been admitted to the ICU at any timepoint during his/her admission. Lastly, we reported the modal (i.e., most frequent) recovery timepoint, again split by whether a patients had been admitted to the ICU or not at any point during admission.

Reviewer 1:

Minor comment: Dear authors, many thanks for your work. Its interesting and globally needed topic. I recommend drawing a diagram to illustrate the method section and data collection.

We thank the reviewer for the appreciation of our work, and have we incorporated their suggestion to include a diagram in the manuscript to explain the work.

Reviewer 2:

The discussion section provides a comprehensive overview of the SCODA score and its potential applications in tracking the disease severity of patients with COVID-19. Here are a few suggestions for improvement:

We are thankful to the reviewer for the compliments and suggestions for improvement of our manuscript. 

1. It would be helpful to briefly explain how the SCODA score was developed, including the rationale for using the 4C Mortality score and incorporating clinical experience. This would give readers a better understanding of the score's foundation and its potential advantages over other existing scores or classifications.

We have added this section (below in italic) to the first paragraph of the discussion section to elaborate on the development of the SCODA score. This part is however kept brief because we also provide the rational for the use of the 4C Mortality score and incorporation of clinical expertise in the methods section. 

(P13, L7 - 10) Alterations of the 4C Mortality score, based on clinical expertise, were needed to reflect the aim of the SCODA score to classify the disease severity at a time point as opposed to the prediction of clinical outcome (e.g., mortality for the 4C Mortality score).

2. While the discussion mentions that there is currently no comprehensive score available for tracking the disease severity of COVID-19 patients over time, it would be beneficial to elaborate on the significance of this gap in knowledge. Discuss how the SCODA score fills this gap and how it can provide valuable insights into disease progression and its correlation with biomarkers.

In this part of the discussion, we explain that the lack of a score that can classify a patients disease severity from day-to-day leads to studies where disease severity is defined very roughly. Either on the initial presentation of the patient (e.g., whether a patient needs to be admitted to the ICU at admission) or based on the outcome (e.g., whether a patient has survived). Specific and longitudinal data on disease severity will allow to identify (subtle) trends in biomarkers which could for example lead to targets for/monitoring of interventions or better prediction models. 

We have added this last part to elaborate on the potential benefits of using the SCODA score in research. 

(P13-14, L26-2) These correlations with biomarkers can subsequently lead to new targets for monitoring of interventions or for developing and updating prediction models. 

3. Further elaborate on the point regarding the potential impact of temporal factors, such as the change in dominant COVID-19 variants, public awareness, and treatment protocols, on the interpretation of the SCODA score measurements. Discuss how these factors may influence the timing of disease severity and recovery, and acknowledge their potential influence on the generalizability of the findings.

We agree that the temporal factors, such as the changing dominant variant, complicate research on COVID-19. For the SCODA score we think these factors will mainly affect the generalisability of the aggregate measures (i.e., peak in disease severity for the group and modal timepoint of recovery) which we quantified for the beat-covid cohort. For the accuracy of the score itself we do not think these temporal factors will have a big impact since the situation where people are vaccinated or there is a less severe dominant variant will result in people getting less sick and this will be reflected by a lower score. Only when new variants result in different disease profiles, e.g., not affecting the airways, which are not captured by the variables in the score, a mismatch might arise between the actual disease severity and the level of the SCODA score. This would warrant adaptation of the score. 

We have added the following text on the potential effects of temporal factors on the performance of the score to the discussion of the manuscript. 

(P14-15 L26-6) So this limitation mainly applies to the question of generalisability of the (aggregated) measurements in our BEAT-COVID cohort. The SCODA score itself will likely not be affected by these temporal factors as the variables included in the score are independent from these factors. For example, when a patient is vaccinated but still gets severely ill with high CRP, the need for mechanical ventilation and a decreased GCS, his/her disease severity will still be accurately quantified despite the vaccination status. Only when new COVID-19 variants lead to different symptom patterns (e.g., severe gastro-intestinal or urinary tract symptoms without concomitant respiratory symptoms) will there be a mismatch in the SCODA score and the actual disease severity.

4. Although the discussion mentions the need for validation and acknowledges potential limitations of the SCODA score, it would be helpful to explicitly state these limitations. Discuss any inherent biases or uncertainties associated with the score, as well as any potential sources of error or confounding factors that may affect its accuracy or applicability.

Since the SCODA score was designed to quantify patients’ disease severity, something for which there is no golden standard, it is difficult to address biases in terms of systematic error as there is no absolute truth against which the results of the score can be compared. Therefore, the main uncertainty in using the SCODA score in other settings is whether the interpretation of disease severity, as quantified by the score, reflects the subjective interpretation of the clinicians. This is why the score should be validated by new researchers and with each successful validation the likelihood that the score reflects the disease severity accurately increases. 

We have added this elaboration to the discussion

(P14, L9-14) This validation is needed because the SCODA score is developed to capture disease severity, something for which there is no golden standard. Therefore, the main uncertainty in using the SCODA score in other settings is whether the interpretation of disease severity, as quantified by the SCODA score, reflects the subjective interpretation of the clinicians. This is why the SCODA score should be validated by new researchers and with each successful validation the likelihood that the score reflects the disease severity accurately increases

---

## [Decision Letter · Decision Letter 1]

24 Aug 2023

Daily disease severity in patients with COVID-19 admitted to the hospital: the SCODA (severity of coronavirus disease assessment) score

PONE-D-23-08857R1

Dear Dr. Smeets,

We’re pleased to inform you that your manuscript has been judged scientifically suitable for publication and will be formally accepted for publication once it meets all outstanding technical requirements.

Kind regards,

Yujiro Matsuishi

Academic Editor

PLOS ONE

Additional Editor Comments (optional):

Dear Dr. Mark Smeets,

I am pleased to inform you that your manuscript has been accepted for publication.  

My comments, and any reviewer comments, are below.   

Your accepted manuscript will now be transferred to our production department.

We appreciate you submitting your manuscript to Plos One. and hope you will consider us again for future submissions.

Kind regards,    

Yujiro Matsuishi

Academic Editor

PLOS ONE

Editor and Reviewer comments:

#Reveiwer 1

All comments have been addressed.

#Reveiwer 2

All comments have been addressed.

Reviewers' comments:

Reviewer's Responses to Questions

**Comments to the Author**

1. If the authors have adequately addressed your comments raised in a previous round of review and you feel that this manuscript is now acceptable for publication, you may indicate that here to bypass the “Comments to the Author” section, enter your conflict of interest statement in the “Confidential to Editor” section, and submit your "Accept" recommendation.

Reviewer #1: All comments have been addressed

Reviewer #2: All comments have been addressed

2. Is the manuscript technically sound, and do the data support the conclusions?

Reviewer #1: Yes

Reviewer #2: Yes

3. Has the statistical analysis been performed appropriately and rigorously? 

Reviewer #1: Yes

Reviewer #2: Yes

4. Have the authors made all data underlying the findings in their manuscript fully available?

Reviewer #1: Yes

Reviewer #2: Yes

5. Is the manuscript presented in an intelligible fashion and written in standard English?

Reviewer #1: Yes

Reviewer #2: Yes

6. Review Comments to the Author

Reviewer #1: Dear Authors,

Many thanks for your response. I have checked the diagram. I hope success for each one of you.

Regards

Reviewer #2: (No Response)

7. PLOS authors have the option to publish the peer review history of their article (what does this mean?). If published, this will include your full peer review and any attached files.

Reviewer #1: No

Reviewer #2: No

---

## [Editor Report · Acceptance letter]

1 Sep 2023

PONE-D-23-08857R1 

Daily disease severity in patients with COVID-19 admitted to the hospital: the SCODA (severity of coronavirus disease assessment) score 

Dear Dr. Smeets:

I'm pleased to inform you that your manuscript has been deemed suitable for publication in PLOS ONE. Congratulations! Your manuscript is now with our production department. 

Kind regards, 

on behalf of

Dr. Yujiro Matsuishi 

Academic Editor

PLOS ONE